# Adaptive Circuit Behavior and Generalization in Mechanistic Interpretability

## Abstract

Mechanistic interpretability aims to understand the inner workings of large neural networks by identifying *circuits*, or minimal subgraphs within the model that implement algorithms responsible for performing specific tasks. These circuits are typically discovered and analyzed using a narrowly defined prompt format. However, given the abilities of large language models (LLMs) to generalize across various prompt formats for the same task, it remains unclear how well these circuits generalize. For instance, it is unclear whether the model's generalization results from reusing the same circuit components, the components behaving differently, or the use of entirely different components. In this paper, we investigate the generality of the indirect object identification (IOI) circuit in GPT-2 small, which is well-studied and believed to implement a simple, interpretable algorithm. We evaluate its performance on prompt variants that challenge the assumptions of this algorithm. Our findings reveal that the circuit generalizes surprisingly well, *reusing* all of its components and mechanisms while only adding additional input edges. Notably, the circuit generalizes even to prompt variants where the original algorithm should fail; we discover a mechanism that explains this which we term *S2 Hacking*. Our findings indicate that circuits within LLMs may be more flexible and general than previously recognized, underscoring the importance of studying circuit generalization to better understand the broader capabilities of these models.

## 1 Introduction

Mechanistic interpretability (Elhage et al., 2021) is an increasingly prominent approach to understanding the inner workings of large neural networks. Much work in this area is dedicated to discovering *circuits*, which are subgraphs within the large model that faithfully represent how the model solves a particular task of interest (Olah et al., 2020), by identifying attention heads and paths with significant causal effects on the model output (Vig et al., 2020; Stolfo et al., 2023; Hanna et al., 2023; Prakash et al., 2024). By studying these circuits, researchers aim to uncover simple, human-interpretable algorithms that explain how the model solves the task.

These circuits are typically analyzed only within the specific prompt format used to extract them. However, modern LLMs can often solve the same task across various prompt formats. This raises important questions about how a circuit generalizes when the prompt format is varied, especially when the full model generalizes. For instance, it is unclear whether the model's generalization results from the reuse of the same circuit components, the components behaving differently, or the use of entirely different components. Evaluating the generality of a circuit can provide a deeper understanding of the circuit's behavior and the range of scenarios in which the circuit serves as a valid explanation for the full model's performance.

The generality of circuits has significant implications for mechanistic interpretability. Since circuits are typically extracted and evaluated using a specific set of prompts, there is no prior expectation for them to generalize. In the worst case, a different prompt format might require a completely different circuit. Ideally, however, the same circuit would solve the task across all prompt variants, providing a general and reliable explanation for how the model performs the task. Figure 1 presents several hypotheses for how a circuit could generalize as the prompt format changes. Given that the goal of

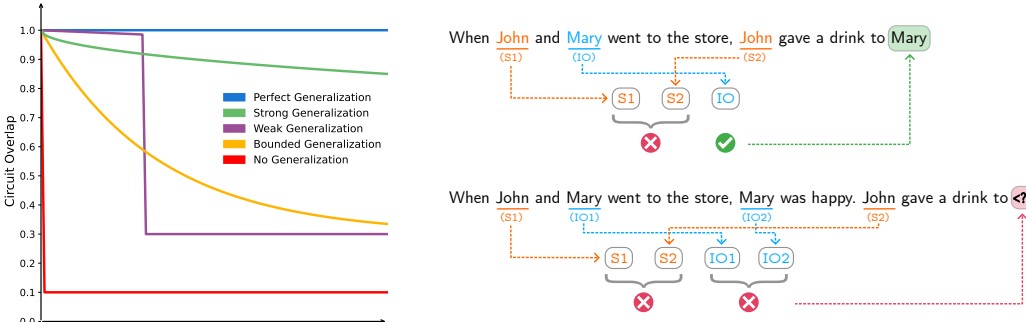

Figure 1: **Left:** Different scenarios for the degree to which a circuit could change as the task format changes. **Right:** The IOI algorithm (top) and the result of applying that algorithm to the DoubleIO prompt variant (bottom) where the subject and indirect object tokens are both duplicated.

mechanistic interpretability is to explain the behavior of the full model, limiting the explanation to a narrow set of prompt formats restricts the insights we can gain about the full model.

*Indirect Object Identification (IOI)* is one of the most well-studied tasks in the mechanistic interpretability literature. The model is given a prompt such as *"When John and Mary went to the store, John gave a drink to _____"*, with the expected answer being "Mary". Wang et al. (2023) discovered a circuit in GPT-2 small that is said to implement a simple three-step algorithm: (1) identify all previous names in the sentence ("Mary", "John", "John"); (2) remove all names that are duplicated ("John"); and (3) output the remaining name ("Mary"). We refer to this as the *IOI algorithm*.

This algorithm appears to be largely agnostic to prompt structure aside from the names, suggesting the circuit may generalize broadly. However, it should clearly fail on prompts where both names are duplicated, since removing all duplicated names leaves no obviously correct answer. For example, consider the prompt *"When Mary and John went to the store, Mary was happy. John gave a drink to _____"*, which breaks the IOI algorithm by adding three words. Yet, GPT-2 small returns the correct answer with $\sim 89\%$ confidence. If the model can be explained by a circuit that implements the IOI algorithm, its ability to generalize to inputs where the algorithm itself predicts failure reveals a clear discrepancy and raises critical questions about the generality of the IOI circuit.

In this paper, we investigate the generality of the IOI circuit. We study its performance and behavior on two prompt variants, *DoubleIO* and *TripleIO*, which are designed to challenge the assumptions of the IOI algorithm. Our key findings are:

(a) The IOI circuit vastly outperforms the full model on prompt variants where the IOI algorithm would completely fail. Despite this, most of the attention heads in the circuit still *retain their functionalities* as specified in Wang et al. (2023).

(b) We discover a mechanism in the IOI circuit, which we call *S2 Hacking*, that explains how the circuit is able to outperform the model on the prompt variants. However, S2 Hacking only appears in these variants and does not arise in the base IOI prompt format.

(c) On discovering new circuits for the prompt variants, we find that they *reuse* all components of the base IOI circuit, only adding edges from additional input tokens. The DoubleIO and TripleIO circuits show $92\%$ and $85\%$ edge overlap respectively, and $100\%$ node overlap in both cases.

These findings reveal that the base IOI circuit generalizes surprisingly well beyond its original task design, reusing many of its heads and paths to handle prompt variants effectively. They also highlight the value of studying circuit generalization, to assess the external validity of the algorithms they are claimed to implement and unveil deeper underlying complexities. Overall, these results represent a significant step towards understanding the more general capabilities of large neural networks and further demonstrate the promise of mechanistic interpretability in achieving this goal.

## 2 BACKGROUND AND RELATED WORK

In this work, we evaluate the circuit for indirect object identification (IOI) in GPT-2 small (Wang et al., 2023) using variants of the prompt format that was originally used to discover the circuit. We refer to the original prompt format as *base IOI*. In the example from Figure 1, "John" and "Mary" are referred to as the *subject* (S) and *indirect object* (IO) tokens respectively, and the IO token is the expected answer. Each base IOI prompt contains one instance of the indirect object token (IO) and two instances of the subject token (S1, S2). We later introduce variants of this prompt that can include multiple instances of the indirect object token (IO1, IO2).

As in Wang et al. (2023), we measure the performance of circuits and the full model using the *logit difference* metric, which is defined as the difference between the log probabilities of the IO token and the S token. A larger positive logit difference indicates that the model predicts the correct token (IO) with higher probability than the incorrect token (S).

The base IOI circuit contains a set of distinct attention head types, each of which performs a specific function in the overall mechanism of the circuit. We include the base IOI circuit diagram in Figure 7, and further details can be found in Wang et al. (2023). We focus on the following head types:

- **Name Mover heads** are responsible for copying the correct token (IO) to the output. These heads are active at the END token and attend to previous names in the sentence, copy the name they attend to, and send it to the output.
- **S-Inhibition heads** ensure that the Name Mover heads focus on the IO token by modifying the queries of the Name Mover heads and suppressing attention to the duplicated tokens (S1, S2). They are active at the END token and attend to S2.
- **Duplicate Token heads** identify tokens that have appeared earlier in the sequence. These heads are active at the duplicate instance of a token (S2), attend to its first occurrence (S1), and output the position of the repeated token.
- **Previous Token heads** copy information from token S1 to the next token (S1+1), the token facilitating the transfer of sequential token information.
- **Induction heads** serve a similar function to Duplicate Token Heads. They are active at the position of the duplicated token (S2) and attend to the token that follows its previous instance (S1+1). Their output is used as a pointer to S1, and to signal that S was duplicated.

A small amount of prior work on circuit discovery has evaluated circuit generalization performance using prompts outside of the format used for their discovery. However, these evaluations are often limited and do not fully investigate how the circuit components and mechanisms behave under these prompt changes. Wang et al. (2023) evaluated the base IOI circuit on adversarial examples similar to the DoubleIO variant we use in this paper, and observed a drop in model performance. We build on this finding by evaluating the circuit itself and explaining the mechanism behind the model's lower yet nontrivial performance. The Greater-Than (Hanna et al., 2023) and Arithmetic (Stolfo et al., 2023) circuits were evaluated on prompt variants and included circuit overlap comparisons, though these studies are mostly qualitative and not explored in further detail. The EAP-IG circuit discovery method (Hanna et al., 2024) was also evaluated using circuit overlap and cross-task faithfulness.

## 3 TESTING GENERALITY OF THE IOI CIRCUIT USING PROMPT VARIANTS

Inferring a circuit from model behavior is a critical step, but it is not sufficient on its own—such a circuit acts as a formal hypothesis about the problem-solving mechanisms of the full model, and this hypothesis should be probed in various ways to determine the circumstances in which it holds. We aim to assess the generality of the base IOI circuit and the range of circumstances in which its core mechanisms persist. To this end, we introduce two *variants* of the base IOI prompt that, according to the original description of the IOI algorithm, should be unsolvable by the base IOI circuit.

In this section, we present three key findings. First, we demonstrate that the unmodified base IOI circuit is unexpectedly capable of sustaining its performance as we systematically vary the nature of the task. Secondly, we demonstrate that the circuit significantly outperforms the model on the variants, showing consistently high logit difference scores while the model performance drops. Finally, we show that most of the attention heads in the circuit behave nearly identically to how they would

on base IOI inputs. These findings suggest that the circuit effectively solves the task exactly as it would on base IOI prompts, even though the necessary conditions for its success, as explained by the IOI algorithm, are not met by the new prompts.

## 3.1 IOI Prompt Variants

The IOI algorithm assumes that the subject (`S`) token is duplicated while the indirect object (`IO`) token is not, and accordingly functions by detecting and suppressing the duplicated token. We can test how much the performance of the base IOI circuit relies on this assumption by manipulating the number of instances of the `IO` token. Our study includes the following range of prompts:

**Base IOI**: When John and Mary went to the store, John gave a drink to ____.

**DoubleIO**: When John and Mary went to the store, Mary was happy. John gave a drink to ____.

**TripleIO**: When John and Mary went to the store, Mary was happy. Mary sat on a bench. John gave a drink to ____.

In the DoubleIO variant, both the `S` and `IO` tokens are duplicated, making it unclear which one the circuit should detect and suppress. In the TripleIO variant, the `IO` token appears three times and the `S` token appears twice. Given the stated logic of the IOI algorithm, where the most frequently duplicated token is suppressed, the circuit should return the `S` token since it now appears most frequently in the prompt.

## 3.2 Base IOI Circuit Performance on Variants

We first evaluate the performance of the full model as well as the base IOI circuit on datasets of 200 prompts generated for each of the variants. Our data generation strategy follows from Wang et al. (2023); we create a set of template sentences for each variant, along with lists of names, places, and objects, and sample from these to construct full sentences.

Table 1 shows the logit difference scores for the full model and the base IOI circuit on each of the variants. Faithfulness is defined as the ratio of the circuit's logit difference to the model's logit difference, which is a measure of how closely the circuit's performance aligns with that of the model.

| Task | Model Logit Difference | Circuit Logit Difference | Faithfulness |
|------|------------------------|--------------------------|--------------|
| Base IOI | 3.484 | 3.119 | 0.895 |
| DoubleIO | 2.118 | 2.722 | 1.285 |
| TripleIO | 1.227 | 3.174 | 2.586 |

Table 1: Base IOI circuit performance on DoubleIO and TripleIO inputs. The circuit maintains high performance while model performance drops, and faithfulness is far from the ideal value of 1.

While we anticipated that the full model might generalize to these new tasks, the base circuit consistently outperforms the model on both DoubleIO and TripleIO variants with near-perfect accuracy. This is particularly surprising because these variants were designed with the explicit purpose of being unsolvable by the circuit if it were to execute its hypothesized algorithm exactly. We also see that the faithfulness scores for the base IOI circuit on the DoubleIO and TripleIO variants is far above 1, since the model performance is lower on the variants while the circuit performance remains consistently high. This indicates that the performance of the base IOI circuit on the prompt variants is not faithful to the full model, and inconsistent with the hypothesized explanation of the circuit.

Wang et al. (2023) also suggested using adversarial examples that are very similar to DoubleIO, though they only evaluate the performance of the full model and do not explore how the circuit performs on the same task. The performance of the full model on the DoubleIO task is still non-trivial; a logit difference of 2.138 indicates that the model predicts `IO` over `S` $\sim 89\%$ of the time.

## 3.3 Head-Level Attention Patterns on Base IOI and Variants

The sharp deviation in performance between the base IOI circuit and the full model raises an important question: *Are the elements of the circuit maintaining the same functions as they had on the base*

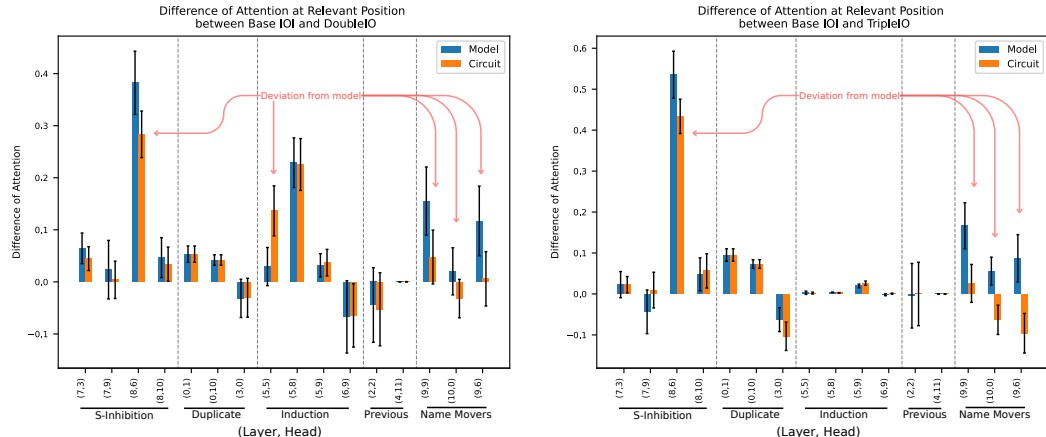

Figure 2: Deviation in attention scores from base IOI inputs to DoubleIO (left) and TripleIO (right) inputs for the base IOI circuit and full model. Nonzero values indicate deviation in behavior due to the change in prompt format. For the circuit, most heads show low deviation ($< 0.1$), particularly the Name Mover heads which are responsible for returning the output. Significant differences between the circuit and model indicate that the base IOI circuit is less faithful on the prompt variants.

*IOI task, or have their functions changed in response to changes in the prompt?* To examine this further, we calculate the average difference between the attention scores of each head in the base IOI circuit for its most relevant token position (specified in Wang et al. (2023)) on the base IOI and prompt variant datasets. Since the full model may respond differently to the prompt variants, we do the same calculations for these heads in the model as well. The results are shown in Figure 2.

We observe minimal deviation in attention scores for most heads, typically within 0.05, with only S-Inhibition Head 8.6 deviating significantly for both variants. We also see deviation in Induction heads 5.5 and 5.8 for DoubleIO, though head 5.8 shows similar deviation in the full model as well. In contrast, the full model exhibits greater deviation in attention scores between base IOI inputs and the variants for several other heads, particularly the Name Mover heads. These heads are responsible for returning the output, which highlights the disparity in performance between the circuit and model.

Overall, these results suggest that most components of the base IOI circuit show lower deviations in attention patterns on base IOI inputs and the prompt variants compared to the full model. We later show in Section 5 that these heads retain their original functionalities from the base IOI circuit.

## 4 S2 HACKING: PERFORMANCE WITHOUT FAITHFULNESS

We observed in the previous section that the base IOI circuit significantly outperforms the full model on the IOI prompt variants, with most of its components maintaining similar attention patterns to base IOI inputs. In this section, we identify the source of the deviations in behavior and performance between the base IOI circuit and the full model.

The basis of the discrepancy between the circuit and the model performance is a mechanism in the base IOI circuit which we term *S2 Hacking*. In the base IOI circuit, the Induction and Duplicate Token heads are primarily active at the `S2` token, which is always the incorrect answer in each of the IOI prompt variants. The outputs of these heads are then used by the S-Inhibition heads to suppress the attention on the `S1` and `S2` tokens. This mechanism is a key factor in how the circuit is able to return the correct (`IO`) token with high probability.

However, evaluating this circuit requires *knocking out* all paths that are not part of the circuit using *mean ablation* (Wang et al., 2023). In the base IOI circuit, the `S2` token is the only input token that has a path from the input tokens to the `END` token that passes through the Duplicate Token and Induction heads. This is because the paths from all other input tokens were shown to have a low causal effect on the model output during the circuit discovery process. As a result, the paths from all other input tokens to these heads are knocked out.

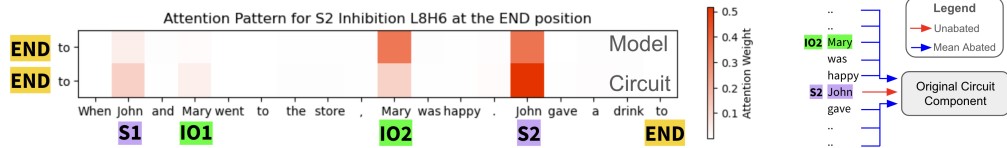

Figure 3: **S2 Hacking in S-Inhibition head 8.6. Left:** Attention pattern at the END position for a DoubleIO prompt. Placing all attention on the S2 token would lead to near-perfect accuracy on the task. Head 8.6 splits attention between IO2 and S2 in the full model, but in the base IOI circuit it focuses primarily on S2. **Right:** Knockout procedure for evaluating circuits, where paths that are not part of the circuit (marked in blue) are mean-ablated out. For head 8.6, the paths from all input tokens other than S2 are knocked out, leading to S2 Hacking.

This knockout procedure effectively points the circuit toward the correct answer every time, since the S2 token is the only input to the Duplicate Token heads that is not knocked out. These heads feed into the Induction and S-Inhibition heads, which are directed to only attend to subject-related tokens, S+1 and S2 respectively. As a result, the S-Inhibition heads always suppress attention on the subject tokens (S1, S2), which pushes the Name Mover heads towards returning an IO token. This mechanism of consistently suppressing the subject tokens and returning the IO token enables the base IOI circuit to solve the task across all of the prompt variants.

We refer to this phenomenon as S2 Hacking. Note that this phenomenon only occurs in the base IOI circuit, as it is a byproduct of the knockout procedure for evaluating the circuit and not actually how the full model solves the task. This is evident from how much the circuit performance deviates from the model performance. Additionally, S2 Hacking is not observed on base IOI inputs since they only have one duplicated name (S), and was discovered by generalizing the prompt format. For the rest of this section, we focus on the DoubleIO prompt variant and demonstrate how S2 Hacking occurs within the base IOI circuit. Additional details on all metrics and experiments are in Appendix A.

## 4.1 METRICS

Based on the head types and their functions specified by Wang et al. (2023), each head is typically characterized by its focus on a specific token or set of tokens. We define the following metrics to compare the attention scores of these specific tokens for each of these head types, to understand the deviations in their behavior between the base IOI circuit and the full model.

$$\text{Confidence ratio} = \frac{Attn(correct)}{Attn(incorrect)} \qquad \text{Functional faithfulness} = \frac{Attn(token)\,in\,circuit}{Attn(token)\,in\,model}$$

The *circuit confidence ratio* for a given head is considered high ($> 1$) if its attention score in the circuit is higher for the correct token than the incorrect token, and the *model confidence ratio* is similarly defined for the head in the full model. If the circuit confidence ratio exceeds that of the model, it suggests that the circuit is more likely to attend to the correct token than the model is.

*Functional faithfulness* scores are used to compare the attention scores between the model and the circuit for each relevant token; we focus particularly on the S2 and IO2 tokens. The ratio is close to 1 if the circuit attends to the token as much as the model does, while a value greater than 1 indicates that the circuit attends to the token more than the model does, demonstrating a difference in behavior for the head. Confidence ratios and functional faithfulness scores for all heads in the circuit are given in Figure 4, where all metrics are plotted with confidence intervals based on 50 samples.

## 4.2 TRACING THE S2 HACKING MECHANISM

Our results demonstrate that the S2 Hacking mechanism is primarily carried out through S-Inhibition head 8.6, Induction heads 5.9 and 5.5, and Duplicate head 3.0. Through this mechanism, we find that all of the Name Mover heads show higher confidence in predicting an IO token over an S token. The remaining heads in the circuit show similar behavior to their performance on base IOI prompts.

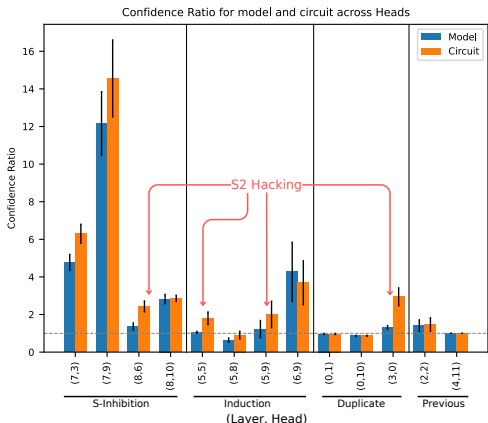 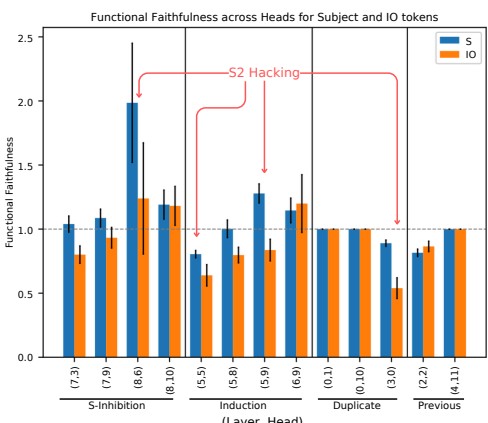

Figure 4: **Left:** Confidence ratios for model and base IOI circuit. S2 Hacking can be seen in heads 8.6, 5.5, 5.9, and 3.0, where confidence ratio is close to 1 for the model but greater than 1 for the circuit. **Right:** Functional faithfulness scores for the `S` and `IO` tokens. The output is more likely to be correct if these heads predict `S`, so high values for the subject token (blue) indicate that the circuit is more confident than the model at predicting the correct answer.

The S2 Hacking mechanism starts from Duplicate head 3.0, as shown by its confidence ratio being close to 1 in the full model but significantly higher than 1 in the base IOI circuit. This indicates that the model places roughly equal attention on the `S2` and `IO2` tokens, but the circuit places more attention on the `S2` token. The functional faithfulness score for the `IO2` token is also low, indicating that the attention on the `IO2` token is significantly lower in the circuit compared to the full model.

The effect of Duplicate head 3.0 cascades down to the Induction heads 5.9 and 5.5. While their confidence ratios are close to 1 in the model, indicating roughly equal attention to both the `S2` and `IO2` tokens, the circuit confidence ratios are significantly higher, suggesting a stronger preference for the `S2` token. Additionally, the functional faithfulness scores show a greater deviation from 1, with significantly lower values for the `IO2` token, indicating that attention to `IO2` is significantly reduced in the circuit compared to the model.

The effects of S2 Hacking on the above heads flow into S-Inhibition head 8.6, which receives input from the Duplicate heads. In this head, the circuit confidence ratio is greater than 2 while the model confidence ratio remains close to 1. Moreover, the functional faithfulness score for the `S2` token is around 2 in the circuit but closer to 1 in the model. These results indicate that the head attends roughly equally to the `S2` and `IO2` tokens in the full model, but in the circuit this head significantly favors the `S2` token. This enables it to effectively direct the Name Mover heads to focus much less on the `S2` token and thereby avoid the incorrect answer.

All the other heads in the circuit do not appear to benefit significantly from S2 Hacking. In particular, the Previous Token heads appear largely unaffected and show very little deviation in behavior between the circuit and the model. While the S-Inhibition heads 7.3 and 7.9 exhibit significantly higher confidence ratios in the circuit compared to the model, their model confidence ratios are already very high. This indicates that they predominantly attend to the `S2` token in both the circuit and the model. These heads suggest how the model can still achieve nontrivial performance on the DoubleIO prompt variant, even if it is not as high as its performance on base IOI prompts.

## 5 HOW DOES GPT-2 SMALL ACTUALLY SOLVE DOUBLEIO AND TRIPLEIO?

Having established that S2 Hacking explains how the base IOI circuit is able to outperform the full model on the DoubleIO and TripleIO variants, we now investigate how the full model actually solves these variants. To do this, we discover a new circuit for each variant using the same patch patching methodology and experimental framework that was used to discover the base IOI circuit (Wang et al., 2023). In addition to explaining how the model solves the DoubleIO and TripleIO variants, these circuits reveal that all components of the base IOI circuit are reused for these variants.

## 5.1 ADDING PATHS FROM INPUT TOKENS

The S2 Hacking mechanism suggests a starting point for discovering DoubleIO and TripleIO circuits. Recall that in the base IOI circuit, S2 is the only input token with outgoing paths to the Duplicate heads, with all paths from the remaining input tokens being ablated out. For the variants, we start with the base IOI circuit and restore some of these paths from other input tokens that were originally ablated out, and see if any of them have a causal effect on the output of the model. The results are shown in Figure 5.

For DoubleIO, we observe that adding paths from the IO2 token brings the circuit's performance closest to the full model, with a normalized faithfulness of 0.77. This is done by adding 10 edges to the base IOI circuit: two for each of the three Duplicate heads and the two Previous Token heads they depend on. In contrast, adding paths from just the IO1 token has little impact on the circuit's performance, while adding paths from other input tokens further degrades its performance. Hence for every path from the S2 token to a Duplicate Token head, the corresponding path from the IO2 token to the same head also has a causal effect on the model output.

A similar pattern emerges in TripleIO, where adding paths from the IO2 and IO3 tokens brings the circuit's performance closest to that of the full model, achieving a normalized faithfulness of 0.79. This requires adding 20 edges to the base IOI circuit: the same 10 edges corresponding to the IO2 token that were added to the DoubleIO circuit, plus 10 edges corresponding to the IO3 token.

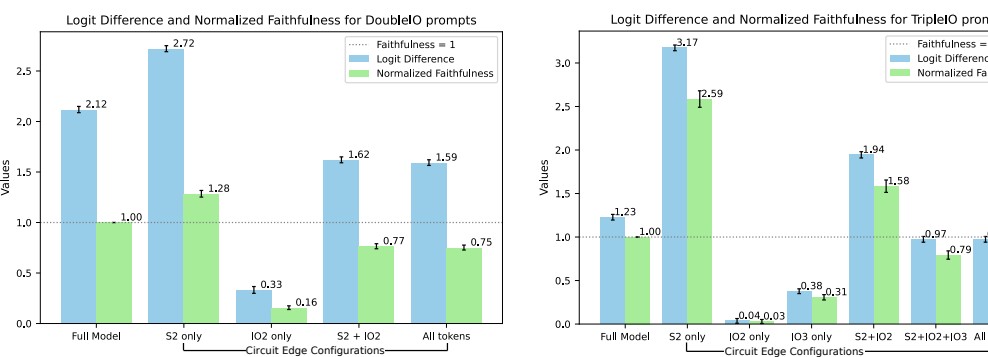

Figure 5: Logit difference and normalized faithfulness for DoubleIO (left) and TripleIO (right) after adding paths to the Duplicate and Previous Token heads from different input tokens. For both variants, the faithfulness is closest to 1 (ideal) when including paths from the input tokens corresponding to duplicated names: S2 and IO2 for DoubleIO, and S2, IO2, and IO3 for TripleIO.

## 5.2 DISCOVERING CIRCUIT REUSE THROUGH PATH PATCHING

We use the methodology of Wang et al. (2023) to discover a circuit for the DoubleIO and TripleIO variants, starting with the identification of Name Mover heads. For each attention head and relevant input token, we compute the direct causal effect of the path that starts from the token and proceeds through the head to the final logit at the END position. This type of token-level causal effect estimation is important for variants like DoubleIO and TripleIO where multiple tokens are duplicated, since each of these duplicates can have paths to different heads.

Surprisingly, we do not observe any significant deviation in results compared to the base IOI circuit. The Name Mover heads from the base IOI circuit are just as causally relevant for the DoubleIO and TripleIO variants, and none of the other heads in the model have a substantial enough direct causal effect score to suggest the addition of a new Name Mover head to the circuit. These results indicate that the Name Mover heads from the base IOI circuit are being reused for both variants.

We observe the same pattern in the S-Inhibition heads, which we refer to more generally as Inhibition heads since they could inhibit either of the duplicated names in the prompt. To identify these heads, we compute the direct causal effects of every head in the model on the queries of the Name Mover heads. We find that all of the S-Inhibition heads from the base IOI circuit have the most significant causal effects, indicating that the DoubleIO and TripleIO circuits are also reusing these heads.

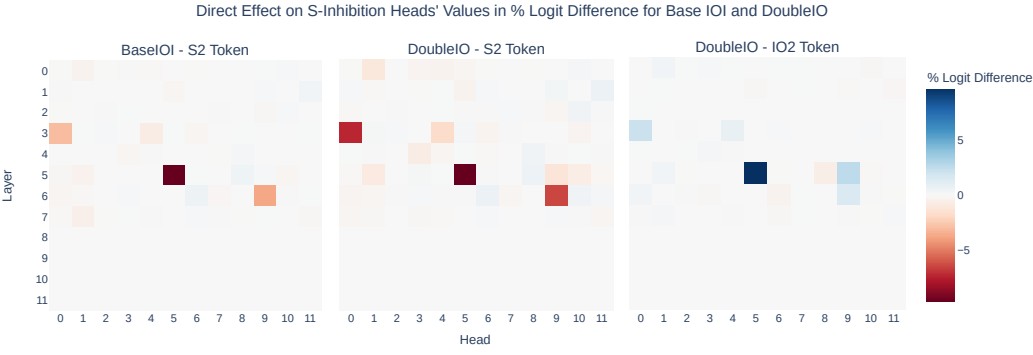

Figure 6: Direct causal effect of all heads in the model on the values of Inhibition heads from the S2 token for base IOI inputs **(left)** and DoubleIO inputs **(middle)**; the heads with the most significant causal effect are similar in both cases. For DoubleIO, these same heads have a significant causal effect in the opposite direction when measured from the IO2 token **(right)**.

Furthermore, the DoubleIO and TripleIO circuits also reuse the Induction, Duplicate, and Previous Token heads from the base IOI circuit, as well as the paths to these heads from the S2 token. To demonstrate this, we compute the causal effect of patching the output of each head in the model from every possible input token to the values of the Inhibition heads.

However, we also find that the paths from the input tokens corresponding to duplicated instances of the IO token also have significant positive causal effect: IO2 for DoubleIO, and IO2 and IO3 for TripleIO. This is consistent with the results in Section 5.1. Since these paths provide input to Inhibition heads, a positive causal effect indicates a negative impact on the overall performance, as increasing the attention on IO tokens leads the Inhibition head to suppress it and reduces the probability of the circuit returning the correct answer. Since the full model has a lower logit difference for these variants, adding these paths from IO tokens increases the circuits' faithfulness.

Overall, the DoubleIO reuses all of the heads and paths from the base IOI circuit while adding 10 more edges corresponding to the IO2 token, and the TripleIO circuit reuses all of the heads and paths from the DoubleIO circuit while adding 10 more edges corresponding to the IO3 token. Table 2 shows the faithfulness of the base IOI, DoubleIO, and TripleIO circuits, and the extent of their overlap. These results align with the *strong generalization hypothesis* from Figure 1 and, to the best of our knowledge, serve as the first demonstration of circuit generalization through circuit reuse.

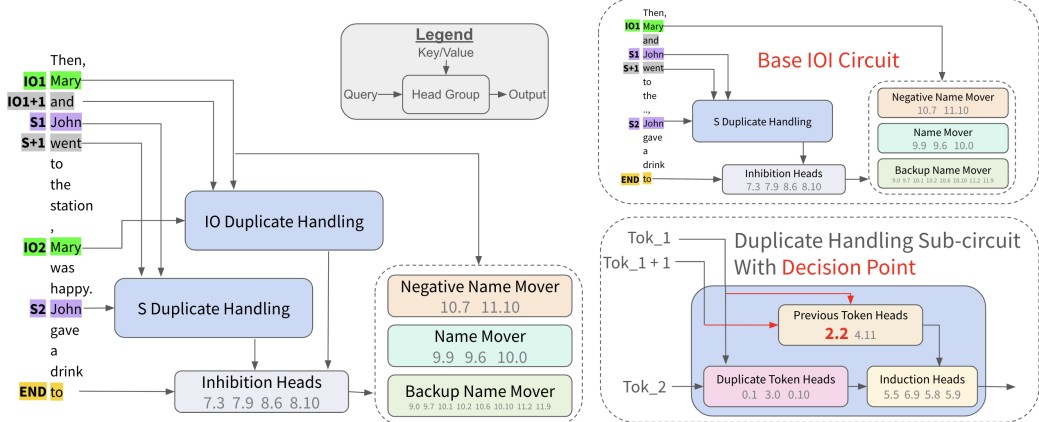

Figure 7: **Left:** Circuit discovered for DoubleIO. All nodes and edges from the base IOI circuit are reused, with additional edges to the IO Duplicate Handling Sub-circuit. **Bottom Right:** Duplicate Handling Sub-circuit, which appears twice in the circuit to deal with the two duplicated tokens (S, IO) in the DoubleIO prompt. **Upper Right:** Base IOI circuit, reproduced from Wang et al. (2023).

| Metric | Base IOI | DoubleIO | TripleIO |
|---|---|---|---|
| # Nodes in circuit | 26 | 26 | 26 |
| # Edges in circuit | 110 | 120 | 130 |
| Node overlap w/ base IOI circuit | 100% | 100% | 100% |
| Edge overlap w/ base IOI circuit | 100% | 91.66% | 84.61% |
| Model Logit Difference | 3.484 | 2.118 | 1.227 |
| Circuit Logit Difference | 3.119 | 1.621 | 0.974 |
| Normalized Faithfulness | 0.895 | 0.765 | 0.778 |

Table 2: Circuit discovery results for DoubleIO and TripleIO. Both circuits reuse all nodes and edges from the base IOI circuit, with DoubleIO adding 10 edges and TripleIO adding 20 edges.

## 5.3 CHOOSING BETWEEN THE DUPLICATED NAMES

The circuit in Figure 7 demonstrates that Inhibition heads now receive information from both the `IO2` and `S2` duplicate tokens in the DoubleIO prompt. This raises a natural question: *How does the model decide which duplicate to suppress to produce the correct answer?* To answer this, we sought to find *decision points*: heads that are most responsible for choosing between the duplicates.

Surprisingly, we find that the order in which names appear in the prompt significantly affects the performance of both the full model and the DoubleIO circuit, with higher performance when the `IO` token comes first. Figure 8 (left) shows the logit differences stratified by whether `S` or `IO` comes first, and the overall logit difference appears to be an average of the two. This suggests that one or more attention heads respond differently depending on the order of the names in the prompt.

We find this behavior in head 2.2, a Previous Token head. This head appears to implement a *"first come, first serve"* mechanism, where it primarily attends to the name that appears first in the prompt, thereby serving as a key decision point in the circuit. As shown in Figure 8 (right), the head frequently assigns high attention to one of the name tokens (`S`, `IO`), based on which appeared first. In prompts where `IO` appeared first, the head attended far more to the `IO+1` token than it did its `S+1` counterpart, and vice versa when `S` appeared first. The full set of experiments is in Appendix B, and a more detailed study of the duplicate suppression mechanism is left to future work.

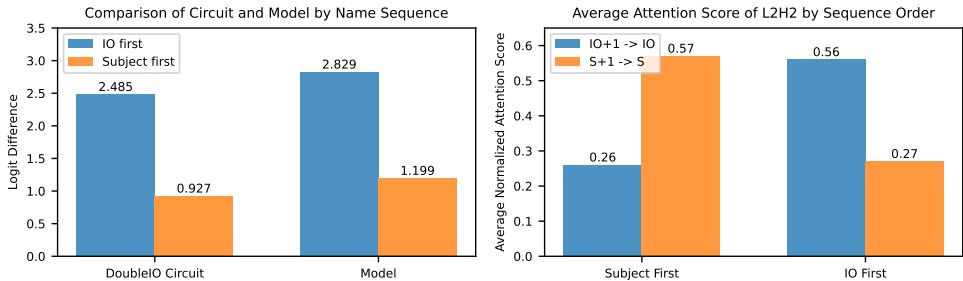

Figure 8: **Left:** Performance of the DoubleIO circuit and full model based on the order of appearance of `S` and `IO` in the prompt. Both perform better when `IO` appears first. **Right:** Average attention scores for head 2.2, which places much more attention on the first name that appears in the prompt.

## 6 CONCLUSION

Our investigation reveals that the IOI circuit in GPT-2 small generalizes more effectively than previously understood, reusing its core components and mechanisms while adapting through minimal structural changes, such as adding input edges. Although the base IOI circuit behaved unexpectedly on the DoubleIO and TripleIO prompt variants, as demonstrated by S2 Hacking, the functionality of its attention heads remained intact. This study of circuit generalization and evaluation on prompt variants ultimately deepened our understanding of how GPT-2 small solves the IOI task, while offering critical insights into the broader capabilities of large neural networks.

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

## A S2 HACKING

In this section, we provide further detail on how each of the attention heads in the base IOI circuit behave under S2 Hacking. The Name Mover heads in the circuit are responsible for returning the output, ideally the `IO` token, and we find that all of these heads attend primarily to the `IO` token. To understand how this occurs, we consider all of the heads that have an incoming edge from the `S2` token: the S-Inhibition, Induction, Duplicate, and Previous Token heads.

### A.1 METRICS

Each head type is defined by a query position ("active" token) and a key position ("attended" token). For DoubleIO prompts, most heads have a decision to focus on tokens leading to the prediction of either `S` or `IO`. For each head type, Wang et al. (2023) specify which tokens to primarily attend to in order to produce the correct answer. We refer to these as the "correct" and "incorrect" tokens for that head type, and they are used in the definition of the confidence ratio in Section 4.1. We specify these "correct" and "incorrect" tokens for each head type below.

For Inhibition heads, which are active at `END`, attending to `S2` suppresses the prediction of the subject token and increases the probability of the Name Mover heads returning the `IO` token, which is the "correct" output. Attending to `IO2`, which suppresses the prediction of the `IO` token, is considered "incorrect". We refer to the attention probability of `S2` at `END` position as the "correct" probability, while the incorrect probability is the attention probability of `IO2` at `END`. Similarly, for

Induction heads we refer to the attention probability of S1+1 at S2 as "correct", and the attention probability of IO1+1 at IO2 as "incorrect". For Duplicate Token heads, we refer to the attention probability of S1 at S1+1 as "correct", and the attention probability of IO1 at IO1+1 as "incorrect". For Previous Token heads, we refer to the attention probability of S1 at S1+1 as "correct", and the attention probability of IO1 at IO1+1 as "incorrect".

## A.2 S-INHIBITION HEADS

The ideal behavior of the S-Inhibition heads on the base IOI prompt is to focus on the duplicated token S2 at the END position. These heads influence the queries of the Name Mover heads, guiding them to reduce their attention on the S2 token. This enables them to focus on the IO token, which is the correct answer. However, since there are two duplicates in the DoubleIO prompt variant, S2 and IO2, it is possible for these heads to attend to both duplicates. The optimal behavior of these heads is to place high attention on the S2 token while maintaining low attention on the IO2 token at the END position.

We see an example of S2 Hacking in Head 8.6, which exhibits a higher confidence ratio in the circuit compared to the model. The model confidence ratio is close to 1 and the circuit confidence ratio is greater than 2, indicating that the head attends roughly equally to the S2 and IO2 tokens in the model but significantly favors the S2 token in the circuit. This enables it to effectively direct the Name Mover heads to focus much less on the S2 token and thereby avoid the incorrect answer.

However, not all S-Inhibition heads rely on S2 Hacking. The confidence ratios for Head 8.10 are very similar in the model and the circuit, in both cases the head attends more to the S2 token. While Heads 7.3 and 7.9 exhibit significantly higher confidence ratios in the circuit compared to the model, the model confidence ratio for these heads are already very high, indicating that they predominantly attend to the S2 tokens regardless. These heads seem to suggest how the model still shows nontrivial performance on the DoubleIO prompt variant, even if not as high as the circuit.

## A.3 INDUCTION HEADS

When an Induction head encounters a duplicate instance of the subject token S2 in the prompt, it attends primarily to the word immediately following the previous instance of the subject (S1+1). There are two duplicates in the DoubleIO prompt variant, so the circuit performance would increase if the induction head placed high attention on the S1+1 token at the S2 position and low attention on the IO1+1 token at the IO2 position. This would ensure that the head continues to prioritize the S2 token.

S2 Hacking is evident in Heads 5.5 and 5.9. Their confidence ratios are close to 1 in the full model, indicating roughly equal attention assigned to both the S2 and IO2 tokens. However, the circuit confidence ratios are significantly higher, suggesting a stronger preference for the S2 token. Additionally, the functional faithfulness scores show higher values for the S2 token compared to the IO2 token, indicating that attention on the S2 token deviates further from the behavior of the full model.

In contrast, Head 6.9 does not appear to rely on S2 Hacking since it already favors the S2 token heavily, as shown by the high confidence ratios on both the model and the circuit and the functional faithfulness scores on the S2 and IO2 tokens are very similar. This is similar to the pattern we observed in the S-Inhibition Heads 7.3 and 7.9. Interestingly, Head 5.8 does not seem to benefit from S2 Hacking and still favors the IO2 token.

## A.4 DUPLICATE HEADS

The function of the Duplicate heads is to focus on the first instance of a name in the prompt when a duplicate instance is encountered. Since the Duplicate heads influence the queries of the Induction heads, it is advantageous for them to attend highly to S1 at the S2 token position while avoiding attending to IO1 at the IO2 token position.

Heads 0.1 and 0.10 demonstrate nearly identical performance between the circuit and the full model, with both attending equally to IO1 and S1. This is evident from the confidence and functional faithfulness scores all being close to 1. In contrast, Head 3.0 shows strong signs of S2 Hacking; while its confidence ratio is close to 1 for the full model, it significantly increases for the circuit. The

functional faithfulness scores further show that the behavior of this head on the S2 token diverges from the full model much more than on the IO2 token. This head shows signs of a decision point where S2 Hacking begins, with its effects cascading down to the Induction and S-Inhibition heads.

### A.5 PREVIOUS TOKEN HEADS

The purpose of a Previous Token head at a given token position is to attend to the token that immediately precedes it in the prompt. In the DoubleIO prompt variant, it would be beneficial for these heads to maintain this functionality at the S1+1 position while minimizing attention on the IO1 token at the IO1+1 position. This is important because the Previous Token heads influence the values of the Induction heads in the base IOI circuit, and we have previously seen how the Induction heads benefit from focusing on the subject token.

Unlike the other head types above, the Previous Token heads appear not to rely very much on the S2 Hacking mechanism. Head 4.11 functions almost identically in both the circuit and the model, with confidence and functional faithfulness scores consistently close to 1. While Head 2.2 slightly benefits from S2 Hacking, the circuit deviates very little from the full model. In both cases, these heads attend more to the S1 token at position S1+1 than they do to the IO1 token at the IO1+1 position, so this head appears to favor the S1 token regardless.

## B IDENTIFYING LAYER 2 HEAD 2 AS THE DECISION POINT

We measure the difference between the direct effect on the S tokens from the Direct Effect on the IO tokens on every relevant head. We selected Previous Token Heads 4.11 and 2.2 as the most likely candidates to be decision points as a result of the aforementioned difference scores. Plotting the attention patterns of each head, however, led us to conclude that Head 2.2 was a better candidate than Head 4.11. This is because while 4.11 was found to have a high causal effect on these tokens, it was only because it was attending highly to both IO and S tokens to a similar degree. Head 2.2 on the other hand, was found to more frequently assign high probably to only one of the two duplicate name tokens, making it the best decision point candidate within the circuit. Plots for the attention patterns of head 4.11 can be found in Figure 9, and the same for head 2.2 in Figure 8.

Previous Token Head 4.11 attends equally to the IO+1 and S+1 tokens, regardless of the order in which the name tokens appear in the prompt. This evidence dissuaded us from considering 4.11 as the decision point where the model would decide which duplicate token to inhibit.

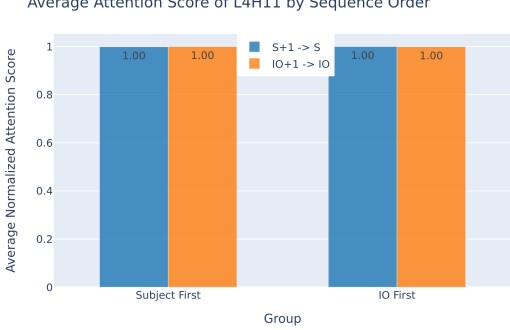

Figure 9: Attention scores of Layer 4 Head 11 indicate that the Previous Token Head attends equally to both the IO and S previous tokens

The attention scores of the Previous Token Head 2.2 show clear changes in the amount of attention assigned to the name tokens dependent upon the order in which they appear in the prompt. When Subjects appear first in the prompt, S+1 tokens have higher attention scores. Conversely, when IOs appear first in the prompt, IO+1 token receives higher attention scores.

The difference between Direct Effect measured on the S tokens and on the IO tokens show high causal effect from the Previous Token Heads 2.2 and 4.11.

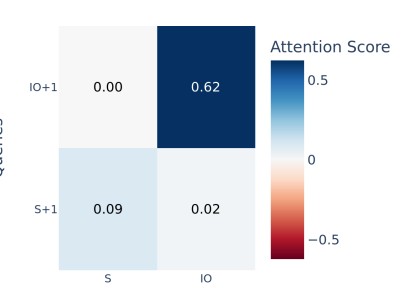 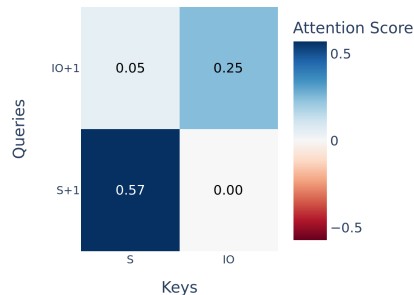

(a) Attention (IO first): Layer 2, Head 2. The attention patterns favor IO when IO appears first.

(b) Attention (Subject first): Layer 2, Head 2. The attention patterns favor Subject when Subject appears first.

Figure 10: Comparison of head 2.2 attention patterns for different input orders.

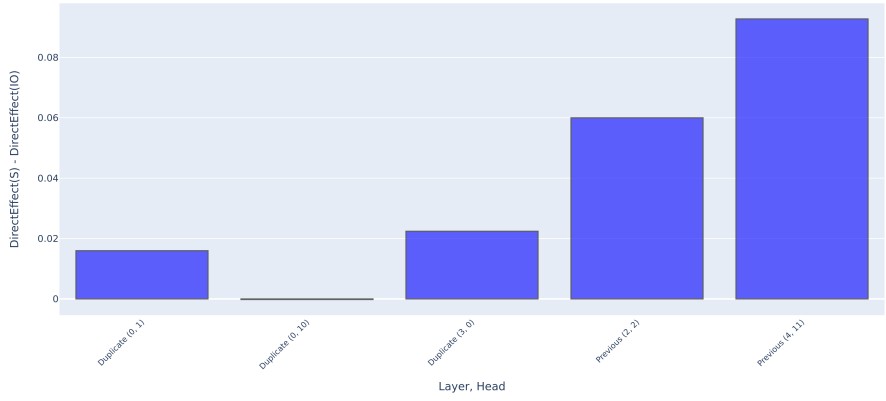

Figure 11: Here we show the difference between the absolute direct effects between the S2 and IO2 edges of the same node. The high value of nodes provide preliminary evidence that they might be a decision point.

## C  PROMPT TEMPLATES

Here we provide the BABA-style prompt templates to generate the DoubleIO and TripleIO datasets and prompts. The ABBA-style format is generated by switching the order of the first two names.

| **BABA Templates - DoubleIO** |
|---|
| Then, [B] and [A] went to the [PLACE], [A] was happy. [B] gave a [OBJECT] to [A] |
| Then, [B] and [A] had a lot of fun at the [PLACE], [A] seemed distracted. [B] gave a [OBJECT] to [A] |
| Then, [B] and [A] were working at the [PLACE], [A] was confused. [B] decided to give a [OBJECT] to [A] |
| Then, [B] and [A] were thinking about going to the [PLACE], [A] was excited. [B] wanted to give a [OBJECT] to [A] |
| Then, [B] and [A] had a long argument, and afterwards [A] smiled. [B] said to [A] |
| After [B] and [A] went to the [PLACE], [A] looked tired. [B] gave a [OBJECT] to [A] |

| |
|---|
| When [B] and [A] got a [OBJECT] at the [PLACE], [A] laughed. [B] decided to give it to [A] |
| When [B] and [A] got a [OBJECT] at the [PLACE], [A] was deep in thought. [B] decided to give the [OBJECT] to [A] |
| While [B] and [A] were working at the [PLACE], [A] was happy. [B] gave a [OBJECT] to [A] |
| While [B] and [A] were commuting to the [PLACE], [A] seemed distracted. [B] gave a [OBJECT] to [A] |
| After the lunch, [B] and [A] went to the [PLACE], [A] was confused. [B] gave a [OBJECT] to [A] |
| Afterwards, [B] and [A] went to the [PLACE], [A] was excited. [B] gave a [OBJECT] to [A] |
| Then, [B] and [A] had a long argument, [A] smiled. Afterwards, [B] said to [A] |
| The [PLACE] [B] and [A] went to had a [OBJECT], [A] looked tired. [B] gave it to [A] |
| Friends [B] and [A] found a [OBJECT] at the [PLACE], [A] laughed. [B] gave it to [A] |

| **BABA Templates - TripleIO** |
|---|
| Then, [B] and [A] went to the [PLACE], [A] sat on a bench. [A] had left their [OBJECT] behind. [B] gave a [OBJECT] to [A] |
| Then, [B] and [A] had a lot of fun at the [PLACE], [A] seemed lost in thought. [A] looked up at the sky. [B] gave a [OBJECT] to [A] |
| Then, [B] and [A] were working at the [PLACE], [A] was laughing softly. [A] glanced at the [OBJECT]. [B] decided to give a [OBJECT] to [A] |
| Then, [B] and [A] were thinking about going to the [PLACE], [A] appeared confused. [A] fumbled with their [OBJECT]. [B] wanted to give a [OBJECT] to [A] |
| Then, [B] and [A] had a long argument, and afterwards [A] smiled. [A] adjusted their [OBJECT]. [B] said to [A] |
| After [B] and [A] went to the [PLACE], [A] yawned. [A] rubbed their eyes. [B] gave a [OBJECT] to [A] |
| When [B] and [A] got a [OBJECT] at the [PLACE]. [A] looked at the ground, [A] sighed deeply. [B] decided to give it to [A] |
| When [B] and [A] got a [OBJECT] at the [PLACE], [A] smiled at the view. [A] seemed relieved. [B] decided to give the [OBJECT] to [A] |
| While [B] and [A] were working at the [PLACE], [A] sat on a bench. [A] had left their [OBJECT] behind. [B] gave a [OBJECT] to [A] |
| While [B] and [A] were commuting to the [PLACE], [A] seemed lost in thought. [A] looked up at the sky. [B] gave a [OBJECT] to [A] |
| After the lunch, [B] and [A] went to the [PLACE], [A] was laughing softly. [A] glanced at the [OBJECT]. [B] gave a [OBJECT] to [A] |
| Afterwards, [B] and [A] went to the [PLACE], [A] appeared confused. [A] fumbled with their [OBJECT]. [B] gave a [OBJECT] to [A] |
| Then, [B] and [A] had a long argument, [A] smiled. [A] adjusted their [OBJECT]. [B] said to [A] |
| The [PLACE] [B] and [A] went to had a [OBJECT], [A] yawned. [A] rubbed their eyes. [B] gave it to [A] |
| Friends [B] and [A] found a [OBJECT] at the [PLACE], [A] looked at the ground. [A] sighed deeply. [B] gave it to [A] |

% Effect of Induction Heads on Inhibition
Head values for S2 and IO2 Indices

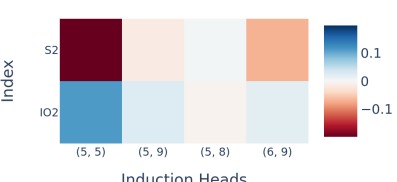

% Effect of Duplicate Token Heads on
Induction Head queries for S2 and IO2

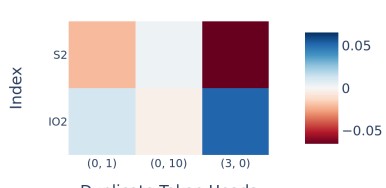

(a) Difference in effects between S2 and IO2
for induction heads.

(b) Difference in effects between S2 and IO2
for duplicate token heads.

Figure 12: Change in direct effect from subject to IO token for induction head and duplicate token heads .

% Effect of Previous Token Heads on
Induction Head keys for S+1 and IO+1

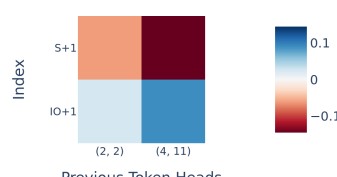

Figure 13: We see that for head 4.11, both Subject and IO edges have high effect, but for 2.2 only Subject has a high effect.

