# OpenReview forum: "Adaptive Circuit Behavior and Generalization in Mechanistic Interpretability"
_ICLR.cc/2025/Conference — Submitted to ICLR 2025_

### Official Review · Reviewer_ERAA · 2024-10-31

**Soundness:** 4
**Presentation:** 4
**Contribution:** 2
**Rating:** 5
**Confidence:** 4

**Summary:**

The paper investigates circuit reuse in language models.
It does so in the context of the IOI task and its circuit.
In particular, it proposes 2 variants to the standard task which follow a similar
structure to the original, and then proceeds to the analyze the circuits/behavior of the
model for these variants.
In these investigations it finds two different things.
1. The standard IOI circuit has really good performance on these tasks despite it being unfaithful.
The authors explain this behavior by a phenomenon they call S2-Hacking. In this phenomenon
the S2 token, via S-inhibition heads + incorrect the mean ablation, ends up artificially pushing
the name mover heads to return the IO token.
2. The circuit for the DoubleIO tasks is very similar to the circuit in the standard IOI task.
The authors discover this by applying standard path-patching techniques to discover the new circuit and comparing it to the original.
This observatin is important as it shows that circuits are "robust"
and are shared in similar enough tasks.

**Strengths:**

- I think the paper was presented in a very clear manner, and the authors did an excellent job of explaining the methodology and their ideas.
- Their methodology seems adequate, as it follows standard practices for circuit discovery.
- I appreciated that the authors investigated the S2-Hacking issue in detail, as I believe this sort of problem can lead to "interpretability illusions" where we might be deceived into thinking a circuit is appropriate when, in reality, it is simply a quirk of the methodology. (I don’t think this fits precisely into that category, as the circuit was quite unfaithful, but it remains an interesting finding.)
- As the authors mention, to the best of my knowledge, this is one of the first careful looks at circuit reuse in LLMs.

**Weaknesses:**

- In my view, the main weakness of this paper is its scope. Although I think the authors did an excellent job presenting their results, I believe that it is necessary to validate the conclusions with more circuits. I understand that several papers only focus on one specific circuit (the IOI paper being a prime example). However, as the techniques presented in the paper are not new and not necessarily its main strength, I believe that more empirical validation should be provided to strengthen the paper's thesis. For example, I would like to see similar analysis (which could be automated) for other circuits, such as Greater-Than [1] or Docstring [2]. I think this would greatly strengthen the paper. Otherwise I think the contributions, despite the excellent presentation and great use of the methodology, might not be enough.



[1] M. Hanna, O. Liu, and A. Variengien. *How Does GPT-2 Compute Greater-Than*. Interpreting Mathematical Abilities in a Pre-Trained Language Model, 2:11, 2023.

[2] S. Heimersheim and J. Janiak. *A Circuit for Python Docstrings in a 4-layer Attention-Only Transformer*. Alignment Forum, 2023. URL: [https://www.alignmentforum.org/posts/u6KXXmKFbXfWzoAXn/a-circuit-for-python-docstrings-in-a-4-layer-attention-only](https://www.alignmentforum.org/posts/u6KXXmKFbXfWzoAXn/a-circuit-for-python-docstrings-in-a-4-layer-attention-only)

**Questions:**

Nitpicks:
-  In paragraph 4 of section 4, should we have S2 rather than S? "These heads then feed into the S-Inhibition heads, which will always suppress attention on the S tokens as a result and push the Name Mover heads toward returning an IO token."
- Have the authors attempted to analyze other circuits in a similar way? If so what have been their findings?
- Are the authors aware of other instances of other "illusions" similar to S2 hacking?

---

> ### Author Response · Authors · 2024-11-23
> **Response to reviewer ERAA**
>
> The reviewer raises an important question about the range of tasks considered in our study. We did explore automated methods for circuit discovery, but found that they consistently overestimated the number of components needed in the circuit. Additionally, these methods are based on attribution scores, which are only an approximation of the true causal effect. For a fair and reliable comparison with the base IOI circuit, which was manually discovered by computing causal effects, we deemed it necessary to use the same methodology in our work and analyze the circuits manually.
>
> That said, the base IOI circuit includes not only a set of model components but also a complete (hypothesized) explanation of its functionality, with clear and falsifiable assumptions. For this reason, we focus on the IOI task and study the generalization of the base IOI circuit.
>
> To support our conclusions, we have added circuit discovery results for the TripleIO variant introduced in Section 3, showing that this circuit also reuses all components of the base IOI circuit and only adds 20 edges. Additionally, we have revised the introduction to highlight different potential scenarios for how a given circuit may generalize as the prompt format changes. While we found that the base IOI circuit generalizes with minimal changes, other circuits, based on different hypotheses, may generalize in different ways. This is an exciting and relevant direction for future research.

---

> > ### Comment · Reviewer_ERAA · 2024-11-23
> >
> > I thank the authors for their comments. Nevertheless, I think that their changes don't address my qualms with the paper. As I mentioned in my original review, I think the paper needs to include more circuits to validate the results. I think that using automatic circuit discovery is one way of doing this but there are other circuits the authors could have used for validation. I appreciate the inclusion of circuit discovery for the TripleIO circuit but I think the task itself needs to be different. Otherwise, it is hard to draw general conclusions about circuit reuse. For these reasons I have decided to retain my score.

---

> > > ### Author Response · Authors · 2024-12-03
> > >
> > > We thank the reviewer for their comments. To clarify, our paper does not aim to make general claims about circuit reuse in GPT-2 small. Instead, we focus on the IOI circuit and investigate how its components generalize. This scoped approach reflects the current limitations of circuit discovery and evaluation methodologies, which are not yet standardized across tasks.
> > >
> > > The suggestion to use additional circuits to make general conclusions is challenging for two reasons. First, existing manually discovered circuits on GPT-2 small use different methodologies for discovery and evaluation, making fair comparisons across circuits difficult. Automated methods, as noted in our previous response, did not yield circuits suitable for such a comparison and are themselves a subject of ongoing evaluation. Second, most of these circuits lack hypothesized algorithms or component-level functionality descriptions, like those provided by the IOI circuit. These are crucial for designing prompt variants that test and falsify these hypotheses.
> > >
> > > Our paper highlights possible generalization behaviors in Figure 1, demonstrating the variability in how circuits might respond to new tasks. We do not claim that results from one circuit can predict the behavior of unrelated circuits.
> > >
> > > We hope this clarifies the scope and focus of our work.

---

### Official Review · Reviewer_8h6V · 2024-11-02

**Soundness:** 3
**Presentation:** 3
**Contribution:** 2
**Rating:** 5
**Confidence:** 4

**Summary:**

The paper evaluates the indirect object identification (IOI) circuit in GPT-2 using variant prompts (DoubelIO and TripleIO). These variants introduce additional objects (IO), thus expected to disrupt the identified circuit’s performance. The authors found that the circuit maintains high performance due to a ‘S2 hacking’ mechanism, where the head defaults to the correct token. The base IOI circuit is largely reused for the variant prompts.

**Strengths:**

1. Originally: Evaluated the IOI circuit using variant prompts, providing new insights into its robustness.
2. Quality: The approach with the DoubleIO variance is motivated and effective.
3. Clarity: The writing is clear and easy to understand.
4. Significance: demonstrated that the IOI circuits are more adaptable than previously understood.

**Weaknesses:**

1. The paper could benefit from a clearer discussion of its motivation and broader context. The authors highlight "circuit analysis as a promising angle.” Is it expected that a "simple, human-interpretable algorithm" exists for any task? So far, the study feels narrowly confined to a specific problem. The authors can discuss how the approach in the paper might generalize to more complex tasks. One potential way to address this is to relate the findings to more general problems, such as in NLP, recognizing the relation between subjects in a sentence.
2. The paper relies heavily on differences in attention scores to evaluate circuit functionality, but it lacks a thorough justification for this choice. Is it possible for the same functionality to be represented by different attention patterns? Alternatively, could similar attention scores result in different interpretations of functionality? A clearer explanation of the link between attention and underlying mechanisms is needed.
3. Some statements lack contextual clarity, such as the "first demonstration of generalization via circuit reuse." Providing concrete context or evidence for each claim would enhance rigor and clarity.

**Questions:**

1. How might the proposed circuit-based approach manage tasks that involve multiple subjects or ambiguous variables? For example, are there instances where tokens other than the IO or S have the highest probability? If so, would the ratio score introduced in the paper sufficiently capture the model’s mechanism in these cases, or would normalizing by the highest predicted token likelihood be necessary?
2. The current approach relies on identifying the functionality of attention heads by using predefined target tokens. Could you elaborate on its adaptability to tasks where reasoning components are less clearly defined, or multiple relevant variables may need to be considered?
3. Minor Point: Please consider reporting variances for all values presented, such as in Figures 7 and 8, to give a more comprehensive view of the data.

---

> ### Author Response · Authors · 2024-11-23
> **Response to reviewer 8h6V**
>
> The reviewer raises important questions about extending this work to tasks with ambiguous variables or multiple subjects. This is a broader challenge in mechanistic interpretability, which mostly focuses on narrow tasks with well-defined answers. We focus on the IOI task defined in Wang et al. 2023, who write *“we chose this task because it is linguistically meaningful and supports an interpretable algorithm”*. Our paper, and mechanistic interpretability in general, aims to develop a scientific understanding of specific mechanisms in large models like GPT-2. The field is at an early stage where it is more feasible to study tasks where such mechanisms can be isolated and understood.
>
> Since circuits are discovered by computing causal effects (Vig et. al 2020), well-defined interventions and causal effect measures are required for a causal effect to be identifiable. For more complex tasks with ambiguous variables, it is likely that mechanisms are distributed across the model, making reliable causal inference difficult or impossible due to unidentifiable causal effects. This remains an open challenge for the field. However, our focus is not on solving this general problem, but rather on a deeper investigation of an existing circuit (Wang et al., 2023\) and its hypothesized algorithm.
>
> The reviewer also highlights the use of differences in attention scores to evaluate circuit functionality. Since the head functionalities in the IOI circuit (Wang et. al 2023) were themselves defined based on attention patterns, changes in attention patterns are a natural signal for identifying deviations from these head functionalities. That said, attention patterns could be influenced by other factors even if functionality remains unchanged, or multiple factors could cancel each other out and result in similar attention patterns, as the reviewer points out. This is why we ultimately rely on *causal effect estimation* in Section 5, particularly using path patching (Wang et. al 2023), to isolate the effect of each model component and determine how the model actually behaves on the task. These results form the basis for our final conclusions regarding the model’s performance on DoubleIO and TripleIO.
>
> Additionally, our explanation for S2 Hacking is based on the mechanisms specified in the original IOI circuit diagram (now reproduced in Fig. 7 of our revised paper) and the mean ablation procedure for knocking out model components, rather than on attention patterns alone. We have revised this section to make this explanation more clear and provide examples. However, we also use differences in attention patterns to provide intuition and illustrate how S2 Hacking affects individual heads in the circuit. We hope that this clarifies the role of attention pattern studies in our overall analysis.
>
> We also thank the reviewer for the suggestion to include variances in the plots related to the circuit discovery experiments. We have added confidence intervals to these plots. We also recomputed all numbers in the tables with 10k samples, and in each case, the standard error was very small (\<0.05).
>
> **References**
> Vig, J., Gehrmann, S., Belinkov, Y., Qian, S., Nevo, D., Singer, Y., & Shieber, S. (2020). Investigating Gender Bias in Language Models using Causal Mediation Analysis. Advances in Neural Information Processing Systems.
> Wang, K. R., Variengien, A., Conmy, A., Shlegeris, B., & Steinhardt, J. (2023). Interpretability in the Wild: a Circuit for Indirect Object Identification in GPT-2 Small. In The Eleventh International Conference on Learning Representations.

---

> ### Comment · Reviewer_8h6V · 2024-11-25
>
> Thank you for the detailed response. I appreciate the clarifications and the additional context. However, the paper still needs to address how the current work could generalize beyond the specific circuit studied. For instance, what are some concrete initial steps toward extending the methods and insights to similar tasks or more complex circuits? Additionally, while the role of attention scores in the analysis has been clarified, when attention scores reliably reflect functionality versus when they might lead to misinterpretations would provide clearer criteria. For these reasons, I will maintain my original score.

---

> > ### Author Response · Authors · 2024-12-03
> >
> > We thank the reviewer for their insightful comments.
> >
> > To clarify, our paper does not aim to make general claims about circuit reuse in GPT-2 small or LLMs in general. In particular, we do not claim that results from one circuit can predict the behavior of unrelated circuits. This scoped approach reflects the current limitations of circuit discovery and evaluation methodologies.
> >
> > Our approach to studying circuit generalization using prompt variants and circuit discovery can generally be extended to other circuits. However, there are practical challenges that can arise when doing so. Most other circuits found in GPT-2 small lack hypothesized algorithms or component-level functionality descriptions, like those provided by the IOI circuit. These are crucial for designing prompt variants that test and falsify these hypotheses. This is one reason why the IOI circuit remains the standard baseline for evaluating circuit analysis methods. Additionally, these circuits are discovered and evaluated using different methodologies, making direct and fair comparisons difficult. These challenges are inherent to the field of mechanistic interpretability, and most works focus on specific tasks rather than making broad claims about model behavior.
> >
> > The functionalities of the IOI circuit components were identified and evaluated by Wang et. al (2023) based on attention patterns, and we acknowledge that such methods have inherent limitations. Misinterpretations due to confounding or evaluation idiosyncrasies are indeed a broader challenge in mechanistic interpretability. For example, we identified the issue of S2 Hacking in our work, and similar challenges have been discussed in other studies (e.g., Miller et al. 2024). Our approach used the circuit discovery method in Wang et al. (2023), but can certainly be extended to future, more robust circuit discovery methods.
> >
> > That said, our paper focuses on investigating the generalization of the IOI circuit rather than addressing the broader limitations of circuit discovery methodologies. We hope this addresses the reviewer’s concerns and clarifies the scope and focus of our work.
> >
> > **References**
> >
> > Wang, K. R., Variengien, A., Conmy, A., Shlegeris, B., & Steinhardt, J. (2023). Interpretability in the Wild: a Circuit for Indirect Object Identification in GPT-2 Small. In The Eleventh International Conference on Learning Representations.
> >
> > Miller, J., Chughtai, B., & Saunders, W. (2024): Transformer Circuit Faithfulness Metrics are not Robust. In First Conference on Language Modeling.

---

### Official Review · Reviewer_pQRe · 2024-11-04

**Soundness:** 2
**Presentation:** 2
**Contribution:** 1
**Rating:** 3
**Confidence:** 5

**Summary:**

This paper explores how IOI circuits respond to new modified prompts specifically crafted to disrupt the original pre-defined circuits' functionality. These variations introduce two new sets of IOI prompts: Double IO and Triple IO, named according to the frequency of IO occurrences within the prompts. The study reveals that, while the circuit structures remain largely consistent, their functionalities exhibit slight variations. As the functionality of nodes in the IOI circuits are mainly discussed under the settings of the original IOI prompt, it is not surprising to see the variations in functionality when prompts are different.

**Strengths:**

The discussions in this paper are comprehensive and closely follow the structure presented in "Interpretability in the Wild." Emphasizing the limitations in estimating the functionality of IOI circuits is crucial for advancing the field of interpretability. Such an emphasis underscores the challenges in understanding how these circuits function across varied prompt structures, particularly when circuit consistency does not guarantee consistent functional behavior.

**Weaknesses:**

- The evaluations on circuits discovered on IOI double and IOI triple is extremely trivial. The logit difference is not even close to the full model. It is hard to be convinced that these circuits are even faithful and accurate to reflect how the model actually perform such tasks. These also diminish the credibility and validity of the findings from later parts. The logit difference of circuit should at least recover 80% of that from the full model to be considered as faithful and accurate.
- There is no empirical evidence to show that the originally designed functionality (algorithm) of IOI would fail on the two new prompts. Though I like the idea, it is hard to convince with no evidence that the designed prompts are able to satisfy the purpose. I suggest to run some simulations.
- Since the circuits are found by hand with path patching, and the logit difference varied significantly from the full model, it is hard to justify if the circuits found are served mainly as to make some specific conclusions in the paper such as 'the circuits are largely reused in different prompt settings'.  Showing KL and logit difference together while being close to the full model would be preferred.

**Questions:**

- What is the KL divergence of the three circuits for IOI, double and triple IO.
- Is the functionality of nodes in this paper mainly focus on the change in logit difference similar to IOI paper. If so, why? How about based on change in overall logit distribution?
- Please refer to weakness section.

---

> ### Author Response · Authors · 2024-11-23
> **Response to reviewer pQRe**
>
> We believe the reviewer may have misinterpreted our presentation of the logit difference and faithfulness scores. The evaluations in Section 3 (Table 1\) are of the *base IOI circuit* when given prompts in DoubleIO or TripleIO format, not of the *DoubleIO or TripleIO circuits*. Our finding is precisely that the base IOI circuit is not faithful to the full model on these variants, and this is the subject of our investigation in this paper.
>
> The reviewer expresses concern that there is “no empirical evidence to show that the originally designed functionality (algorithm) of IOI would fail on the two new prompts”. We considered this point to be self-evident given the simplicity of the explicitly stated algorithm: (1) Identify all previous names in the sentence (”Mary”, ”John”, ”John”); (2) remove all names that are duplicated (”John”); and (3) output the remaining name (“Mary”). In DoubleIO and TripleIO, all names are duplicated, thus all names would be removed, leaving no output.
>
> The reviewer also asked about the KL divergence between the circuits and the model. We report these below for the base IOI circuit from Wang et. al 2023, as well as our DoubleIO and TripleIO circuits. The alignment in overall logit distribution is roughly similar for all the circuits.
>
> | Circuit | KL Divergence |
> | :---- | :---- |
> | Base IOI | 0.4799 |
> | DoubleIO | 0.5374 |
> | TripleIO | 0.5577 |
>
> **References**
> Wang, K. R., Variengien, A., Conmy, A., Shlegeris, B., & Steinhardt, J. (2023). Interpretability in the Wild: a Circuit for Indirect Object Identification in GPT-2 Small. In The Eleventh International Conference on Learning Representations.

---

> ### Comment · Reviewer_pQRe · 2024-11-24
>
> Thank you for your response. I appreciate the time that the authors took for rebuttal.
>
> > We believe the reviewer may have misinterpreted our presentation of the logit difference and faithfulness scores. The evaluations in Section 3 (Table 1) are of the base IOI circuit when given prompts in DoubleIO or TripleIO format, not of the DoubleIO or TripleIO circuits. Our finding is precisely that the base IOI circuit is not faithful to the full model on these variants, and this is the subject of our investigation in this paper.
>
> Thanks for the correction. You are right that I misinterpreted this table.
>
> > The reviewer expresses concern that there is “no empirical evidence to show that the originally designed functionality (algorithm) of IOI would fail on the two new prompts”. We considered this point to be self-evident given the simplicity of the explicitly stated algorithm: (1) Identify all previous names in the sentence (”Mary”, ”John”, ”John”); (2) remove all names that are duplicated (”John”); and (3) output the remaining name (“Mary”). In DoubleIO and TripleIO, all names are duplicated, thus all names would be removed, leaving no output.
>
> I am not really convinced that given the previous structure of the IOI algorithm which is not only having the above three mentioned functions would completely fail the double/ triple IOI.
>
> > The reviewer also asked about the KL divergence between the circuits and the model. We report these below for the base IOI circuit from Wang et. al 2023, as well as our DoubleIO and TripleIO circuits. The alignment in overall logit distribution is roughly similar for all the circuits.
>
> Given the KL divergence, it seems likely that the circuits may have been over-pruned. A KL divergence above 0.3 suggests a decent amount of shift from the original output distribution of the full model. This raises questions about whether the pruned circuits can accurately or faithfully reflect the full model's behavior on this task. Because of this, there remains some skepticism about whether the circuits identified by the authors were chosen in a way that specifically supports certain conclusions presented in the paper.
>
> Overall, my concern unfortunately still remains, and I will keep my current ratings toward this paper.

---

> > ### Author Response · Authors · 2024-12-03
> >
> > We appreciate the reviewer’s feedback, and address their concerns below.
> >
> > The IOI algorithm referenced in our paper is quoted verbatim from the beginning of Section 3 of Wang et al. (2023). For clarity, we reproduce it here:
> >
> > *"The following human-interpretable algorithm suffices to perform this task:*
> >
> > *1\. Identify all previous names in the sentence (Mary, John, John).*
> > *2\. Remove all names that are duplicated (in the example above: John).*
> > *3\. Output the remaining name.*
> >
> > *Below we present a circuit that we claim implements this functionality. Our circuit contains three major classes of heads, corresponding to the three steps of the algorithm above."*
> >
> > As explained in our previous response, this algorithm cannot handle DoubleIO and TripleIO inputs because the removal of duplicated names does not leave a single remaining name, and the subject token is no longer the most frequently duplicated name in the input prompt. Thus, the reviewer’s assertion that the IOI algorithm could handle these variants is inconsistent with its original description in Wang et al. (2023).
> >
> > Regarding KL divergence, the base IOI circuit described in Wang et al. (2023) shows similar KL divergence values to our DoubleIO and TripleIO circuits, which were derived and evaluated using the same methodology. Conmy et. al (2023) also report a KL divergence of 0.44 for the base IOI circuit, which exceeds the threshold of 0.3 specified by the reviewer.
> >
> > It remains unclear why the base IOI circuit obtains such KL divergence numbers, and we are not aware of any work claiming that the base IOI circuit is over-pruned based on this. Heimersheim et. al (2024) note that the KL divergence metric *“penalises any deviation (positive or negative), at the cost of also tracking lots of variation we may not care about”.* Additionally, Conmy et. al (2023) show that KL divergence can vary significantly based on the task, e.g. 0.078 for Greater-Than vs. 1.1 for Docstring. This is an open problem in circuit discovery algorithms, though it is not the focus of our paper.
> >
> > That said, our approach can easily be extended to future circuit discovery methods, which may yield different KL divergence results for both the original IOI circuit as well as the DoubleIO and TripleIO variants. We hope this clarifies the methodology used in our study and addresses the reviewer’s concerns.
> >
> > **References**
> >
> > Wang, K. R., Variengien, A., Conmy, A., Shlegeris, B., & Steinhardt, J. (2023). Interpretability in the Wild: a Circuit for Indirect Object Identification in GPT-2 Small. In The Eleventh International Conference on Learning Representations.
> >
> > Heimersheim, S., & Nanda, N. (2024). How to use and interpret activation patching. arXiv preprint arXiv:2404.15255.
> >
> > Conmy, A., Mavor-Parker, A., Lynch, A., Heimersheim, S., & Garriga-Alonso, A. (2023). Towards automated circuit discovery for mechanistic interpretability. Advances in Neural Information Processing Systems, 36, 16318-16352.

---

> ### Comment · Reviewer_pQRe · 2024-12-03
>
> Thank you for the response.
> > It remains unclear why the base IOI circuit obtains such KL divergence numbers, and we are not aware of any work claiming that the base IOI circuit is over-pruned based on this. Heimersheim et. al (2024) note that the KL divergence metric “penalises any deviation (positive or negative), at the cost of also tracking lots of variation we may not care about”. Additionally, Conmy et. al (2023) show that KL divergence can vary significantly based on the task, e.g. 0.078 for Greater-Than vs. 1.1 for Docstring. This is an open problem in circuit discovery algorithms, though it is not the focus of our paper.
>
> Simply because there is no previous work mentioned such problem doesn't mean this problem does not exist. Again, it is hard to prove that the specific circuits the authors pruned are mainly served as making specific statement in the paper (confirmation bias). It is true that for different tasks usually simpler tasks, a small circuit for GT could achieve very well K-L divergence. For IOI, it is generally requires more components to compute would result a larger circuits. **However, again, given the estimation from KL divergence, it is not convincing that the pruned circuits are accurately reflecting the model's actual performance.**
>
> > That said, our approach can easily be extended to future circuit discovery methods, which may yield different KL divergence results for both the original IOI circuit as well as the DoubleIO and TripleIO variants.
>
> I don't see how it relates to future circuit discovery methods. This sounds like a very big overstatement. By adding doubleIO or tripleIO? Please specify.
>
> Moreover, I also agree with the first concern from the re-review by reviewer **CEH6**.
>
> Overall, my concerns are not solved, and I will remain my score.

---

### Official Review · Reviewer_CEH6 · 2024-11-04

**Soundness:** 1
**Presentation:** 1
**Contribution:** 2
**Rating:** 3
**Confidence:** 4

**Summary:**

The authors study the IOI circuit introduced by Wang et al (2023). [[1]](https://arxiv.org/pdf/2211.00593#page=4.37) They notice that the hypothesized algorithm implemented by the circuit would not work in cases where the IO token appears twice in the prompt (DoubleIO prompt). However, they note that the model and circuit can in fact predict the correct token on the DoubleIO prompt.

The authors hypothesize the circuit is able to perform well on the DoubleIO prompt by "S2-hacking". That is, having the Induction, Previous Token and Duplicate Token heads pay extra attention to the second instance of the S2 token, thereby causing the S-inhibition heads to suppress the prediction of this token and thus predict the IO token more strongly.

They further identify a circuit for the DoubleIO prompt using activation patching and find that it has significant overlap with the original IOI circuit and reuses many of the same components.

**Strengths:**

Evaluating on prompt variants intended to test specific aspects of the operation of the circuits is a good idea.

It is a useful finding that the IOI circuit components are often reused in the DoubleIO prompt formats.

The authors present some interesting experimental results demonstrating the similarity of the functions of the attention heads in the IOI circuit between the IOI and DoubleIO prompts.

**Weaknesses:**

- The experimental reports are lacking in many details about the experimental methodology, making it difficult to be confident that the claims are robust.
- The explanations throughout the paper should be clearer to fully communicate the ideas and experiments of the authors.
- The S2 hacking hypothesis is quite vague and the author do not present any deep understanding that would explain the mechanisms by which certain attention heads pay extra attention to the S2 token.
- In the experiments on the DoubleIO and other prompt variations, it is unclear at which token positions paths are being ablated, as this is unspecified by the original circuit.
- The authors write: “Given the algorithm inferred from the IOI circuit, it is clear that the full model should completely fail on this task”. However this is a misunderstanding of the original work. The IOI circuit was discovered using mean ablations that keep most of the prompt intact. Therefore Wang et al. don’t expect it generalize to different prompt formats.
- The authors write “In the base IOI circuit, the Induction, Duplicate Token, and Previous Token heads primarily attend to the S2 token” this is incorrect according to Section 3 of Wang et al., 2023. These heads are _active_ at the S2 token, but do not primarily attend to it.
- The authors write: "The proposed IOI circuit is shown to perform very well while still being faithful to the full model" In fact, the IOI circuit is known to have severe limitations, as shown in concurrent work by Miller et al. (2024) [[2]](https://arxiv.org/abs/2407.08734).

Nitpicks:
- In Figure 2, it is not clear what Head 1, 2, 3 and 4 refer to.
- The paper should include Figure 2 from Wang et al. 2023 [[1]](https://arxiv.org/pdf/2211.00593#page=4.37) to make it easier to follow discussions about the circuit.

**Questions:**

The authors write: “since the S2 token is the only input with a path to the END token through the Induction, Duplicate Token, and Previous Token heads, these heads do not need to attend to any of the other input tokens” I don't understand what exactly this means. Why do the other heads not need to attend to any other input tokens?

---

> ### Author Response · Authors · 2024-11-23
> **Response to reviewer CEH6**
>
> The reviewer states that Wang et al. do not evaluate their circuit on other prompts and therefore do not expect it to generalize to other formats. However, our findings indicate that both the model and the circuit demonstrate generalization to variants of the base IOI prompt format, which suggests a clear and unexplained discrepancy if Wang et al. did not expect such generalization. Investigating this discrepancy is a key goal of our paper. Additionally, we note that Section 4.4 of Wang et al. includes an evaluation on a variant similar to DoubleIO and shows that the full model generalizes, though they leave the explanation of this behavior to future work.
>
> The reviewer raises questions regarding the token positions where paths are ablated as well as the experimental methodology. When evaluating the base IOI circuit on variants such as DoubleIO, the token positions where paths are ablated are exactly the same as the paths ablated in the original IOI paper (Wang et. al 2023\) — we replicated their methodology using their code. The full set of paths that are not ablated out can be seen in the circuit diagram (Figure 2\) of Wang et. al 2023, which we have now reproduced in the main paper alongside the DoubleIO circuit for reference.
>
> The reviewer also asked why the Induction, Duplicate Token, and Previous Token heads do not need to attend to any other input tokens; this is because the paths from all of these input tokens are ablated out. Only the S2 token has a direct (unablated) path to the Duplicate Token heads, which then feed into the Induction heads. The Induction and S-Inhibition heads also primarily attend to the S2 token, though their mechanism also depends on the Previous Token heads which technically attend to the S1+1 token. In all cases, input tokens relevant to the indirect object (IO1, IO1+1, IO2) are ignored. We have updated the section on S2 Hacking to improve clarity and add examples, with a more detailed explanation of how specific heads attend to specific input tokens in Appendix A.
>
> We are confused by the reviewer’s comment that we “do not present any deep understanding that would explain the mechanisms by which certain attention heads pay extra attention to the S2 token”. The full explanation for this mechanism is specified in the paper: it occurs because the paths from all the other input tokens to those heads are ablated out. Section 4 expands on this, showing which heads are the most involved in S2 Hacking (3.0, 5.7, 5.9, 8.6) and their behavior (attending only to the S2 token). To our knowledge, this is a complete description of S2 Hacking. For additional clarity, we have added an example showing the attention patterns that result from S2 Hacking at the beginning of Section 4\.
>
> We also thank the reviewer for pointing us to concurrent work (Miller et. al 2024\) which highlights the limitations in the ablation methodology in Wang et. al (2023). As our paper shows, S2 Hacking is also a pathology that arises from their mean ablation method. We would like to clarify that the proposed IOI circuit is *claimed* to perform very well while still being faithful to the full model. Our work, as well as other studies like Miller et. al 2024, further investigates this claim. We have incorporated this clarification into the introduction of the paper.
>
> **References**
> Wang, K. R., Variengien, A., Conmy, A., Shlegeris, B., & Steinhardt, J. (2023). Interpretability in the Wild: a Circuit for Indirect Object Identification in GPT-2 Small. In The Eleventh International Conference on Learning Representations.
>
> Miller, J., Chughtai, B., & Saunders, W. (2024): Transformer Circuit Faithfulness Metrics are not Robust. In First Conference on Language Modeling.

---

> ### Comment · Reviewer_CEH6 · 2024-11-24
> **Re-review**
>
> Thanks to the authors for their responses and for significantly improving the clarity of the paper. I did not properly understand the paper in my initial review, and so I have fully re-read the submission. Here is my new review:
>
> **Summary:**
> The authors study the IOI circuit introduced by Wang et al (2023). [1] They note that the hypothesized algorithm implemented by the circuit would not work in cases where the IO token appears twice in the prompt (DoubleIO prompt). However, empirically the model and circuit can in fact predict the correct token on the DoubleIO prompt (although with lower confidence).
>
> The authors find the that the original circuit, when directly transplanted to the DoubleIO prompt performs significantly better than the model. This is because the edges from IO tokens to the Duplicate Handling heads which connect to the Inhibition Heads are not part of this circuit, and therefore the negative name movers do not attend to the IO tokens (which in this case improves the model performance, but is not faithful to the original model). The authors call this "S2-hacking".
>
> They further identify a circuit for the DoubleIO prompt using activation patching and find that it has significant overlap with the original IOI circuit and reuses many of the same components.
>
> **Strengths:**
> Evaluating on prompt variants intended to test specific aspects of the operation of the circuits is a good idea.
>
> It is a useful to investigate how circuits discovered on highly specific prompt formats generalize to other distributions.
>
> **Weaknesses:**
>
> I have two primary concerns:
>
> 1. The results do not seem highly significant. Overall this feels more appropriate to be a workshop paper than a main conference paper. For me the primary takeaway from the S2-hacking result is "be very careful with your ablation method - naively transplanting a circuit to a different prompt with the same token positions won't work." This feels like a fairly minor methodological point. The findings about component reuse are more significant (if they are indeed correct), but not enough to justify a full conference submission.
>
> 2. I am concerned that the experimental results will not hold up to deeper scrutiny. The authors appear to have only performed path patching for _circuit discovery_. Wang et al. perform a much more rigorous _circuit evaluation_ of faithfulness, completeness and minimality by patching the entire circuit at once. And their reported results were fairly shaky (there is great variance between individual prompts). More importantly, even Wang et al.'s more thorough evaluation has not held up very well to deeper scrutiny by Miller et al! So I do not feel confident that the results of this work would not show similar or greater flaws if deeply examined.
>
> Minor points:
>  - “As established in the previous section, the base IOI circuit significantly outperforms the full model on the IOI prompt variants without any changes in the functions of its components” This statement seems not fully justified. Three heads have an attention score difference greater than 0.1, and it is unclear what downstream effects this might have.
> - In Figure 2, the authors should state clearly if the attention score difference is the difference of the sum or mean attention value.
> - “In this section, we identify the source of the deviations in behavior and functionality between the base IOI circuit” typo
> - “In the base IOI circuit, the Induction and Duplicate Token heads primarily attend to the S2 token” as mentioned before, according to Wang et al, these heads are active at the S2 token, but they primarily attend to S1.
>
> **Questions:**
>  - Can you explain the discrepancy between the model logit difference in DoubleIO in Table 1 of your paper and Figure 8 of Wang et al?
>  - “The circuit confidence ratio for a given head is considered high (> 1) if its attention score in the circuit is higher for the correct token than the incorrect token” What is the “correct” token here? Is this the correct completion to the prompt? In which case, is it the sum of all instances of that token in the prompt?
>
> After these considerations, I am keeping my score at a 3.

---

> ### Author Response · Authors · 2024-12-03
>
> We thank the reviewer for their thoughtful re-evaluation of our paper and insightful comments. We have updated the paper to address each of the minor points they mentioned, and we answer their questions below.
>
> The DoubleIO prompt in our paper and the prompt in Figure 8 of Wang et al. induce different output distributions due to differences in how they trigger the *repeated name penalty* (Gordon et. al 1993). Wang et al.’s prompts use short sentences separated by periods, which we found to significantly raise the probability of the model returning a pronoun (e.g. “him”) instead of a name (e.g. “John”). This leads to a lower logit difference score since it only compares the probabilities of the names, but this was not investigated in Wang et al. (2023). To help mitigate this problem, our prompt format instead separates the repeated instances of names using a comma, which is more likely to be in-distribution for the model. We suspect this change is responsible for the deviation in logit differences between the two prompts.
>
> We also thank the reviewer for highlighting the need to clarify the “correct” and “incorrect” tokens in the circuit confidence ratio, and we have updated Appendix A.1 to include definitions for these tokens for each head type. In short, Wang et al. (2023) specifies the tokens at which each head is active (query position) and which tokens it attends to (key position) such that the circuit returns the correct name. We refer to these as the “correct” tokens to attend to. The “incorrect” token is the analogous token where S is replaced by IO and vice-versa, such that the circuit returns the incorrect name. For example, when calculating confidence ratios for an Inhibition head we refer to S2 at the END position as the “correct” token and IO2 at the END position as the “incorrect” token.
>
> Regarding the minor points: we have fixed the typos mentioned and clarified that Figure 2 shows the average difference in attention scores. The reviewer also correctly pointed out that a line following Section 3 was not fully justified; the functionality of each head in the circuit is established in Section 5 of the paper and not Section 3\. We have updated the paper to clarify this.
>
> We recognize that concurrent work by Miller et. al (2024) has raised concerns regarding the methodology of Wang et al. (2023). Indeed, we discover variations in performance between the sequence of IO and S tokens in Section 5.3 of our paper, similar to those described in Figure 8 of Miller et al. (2024). However, our studies of circuit generalization and reuse were designed to test their circuit and hypothesized algorithm, and we believe using their approach ensures a fair comparison. The analysis in our paper can certainly be extended to include more robust methods for circuit discovery, which may even result in a new base IOI circuit. However, these are not the focus of the current work.
>
> Our analysis of circuit generalization is based on faithfulness, and this is also the primary metric studied by Miller et al. (2024) for the IOI circuit. As the reviewer pointed out, calculating these scores requires patching the entire circuit at once, and this is exactly what we did in our experiments.
>
> We hope this addresses the reviewer’s concerns and clarifies the methodology used in our study, and we thank the reviewer for their valuable feedback.
>
> **References**
>
> Gordon, P. C., Grosz, B. J., & Gilliom, L. A. (1993). Pronouns, names, and the centering of attention in discourse. Cognitive Science, 17(3), 311–347.
>
> Wang, K. R., Variengien, A., Conmy, A., Shlegeris, B., & Steinhardt, J. (2023). Interpretability in the Wild: a Circuit for Indirect Object Identification in GPT-2 Small. In The Eleventh International Conference on Learning Representations.
>
> Miller, J., Chughtai, B., & Saunders, W. (2024): Transformer Circuit Faithfulness Metrics are not Robust. In First Conference on Language Modeling.

---

> > ### Comment · Reviewer_CEH6 · 2024-12-03
> >
> > I thank the authors these updates and replies.
> >
> > Please could the authors clarify exactly which experiments involved patching the entire circuit and the precise methodology used. For example, did they patch the circuit or the complement of the circuit? Did they perform patch nodes or edges? What was the type of ablation used? Did they patch all token positions or specific token positions?
> >
> > Given that the authors mentioned that they reused code from Wang et al., I would expect the answers are: they patched the nodes in the complement of the circuit, with mean ablation at specific token positions.

---

> > > ### Author Response · Authors · 2024-12-04
> > > **Clarifying the patching methodology**
> > >
> > > We thank the reviewer for their reply. As they noted, we used the codebase from the original IOI paper (Wang et al. 2023) for our experiments. As such, we mean ablate nodes in the complement of the circuit during evaluation, at specific token positions. We used this approach to produce all the faithfulness scores in the paper, including the results in Tables 1 and 2 and Figure 5. We hope this clarifies the methodology used in our work.

---

### Author Response · Authors · 2024-11-23
**Overall Response**

We thank all the reviewers for providing such useful feedback. The reviews make clear that our submitted draft could have highlighted the key points more clearly: (1) Almost no work in MechInterp has examined circuit generality; (2) generality can be evaluated along specific dimensions by varying the prompts; and (3) systematically evaluating circuit generality can produce extremely useful findings and an increased understanding of discovered circuits.

We have produced a revised draft that more clearly highlights these more general conclusions of the paper, along with the following:

* Including circuit discovery results for an additional variant of the IOI task (TripleIO). We previously showed that the DoubleIO circuit reuses all components of the base IOI circuit, and our new results confirm that the TripleIO circuit similarly reuses all components of the base IOI circuit.
* Providing examples and a clearer explanation of S2 Hacking, one of the phenomena that our probe of circuit generality revealed, in Section 4 and Appendix A.
* Clarifying that we replicate the experimental methodology from the original IOI paper (Wang et. al 2023), and reproducing their circuit diagram for reference.
* Adding confidence intervals to the circuit discovery plots and logit difference numbers. Each was recomputed with 10k samples, and the standard errors were very small (<0.05).

With these revisions, we hope the paper presents a clearer picture of circuit generalization as a subject of study, and a stronger case for the generality of the IOI circuit. Below we respond to specific comments and point to specific sections of the paper that we have updated in response to reviewer comments.

**References**

Wang, K. R., Variengien, A., Conmy, A., Shlegeris, B., & Steinhardt, J. (2023). Interpretability in the Wild: a Circuit for Indirect Object Identification in GPT-2 Small. In The Eleventh International Conference on Learning Representations.

---

### Meta-Review · Area_Chair_L5dv · 2024-12-21

**Metareview:**

Thank you for your submission to ICLR. This paper investigates the Indirect Object Identification (IOI) circuit in GPT-2 small, and assesses its behavior under prompt variations (DoubelIO and TripleIO) that intend to disrupt the IOI circuit. Finding that this model still performs well under these prompt variations, the authors explain this behavior through the phenomena of S2 hacking.

Reviewers appreciated the clarity of presentation and insights provided by this paper, and agree that there is merit in studying the IOI circuit. However, there were concerns about the significance of the findings, the scope and extent of the empirical results, and whether the results generalize to circuits beyond the specific circuit being studied. I encourage the authors to carefully consider and incorporate these comments from reviewers upon resubmission.

**Additional Comments On Reviewer Discussion:**

After the author rebuttal, the reviewers had a healthy discussion, taking the authors’ responses into account. One reviewer realized that they had certain misunderstandings and wrote a full re-review of the paper. However, on the whole, the reviewers remained unconvinced about their concerns.

---

### Decision · Program_Chairs · 2025-01-22

Reject